# Clinical Outcome of Arthroscopic Repair for Isolated Meniscus Tear in Athletes

**DOI:** 10.3390/ijerph20065088

**Published:** 2023-03-14

**Authors:** Goran Vrgoč, Filip Vuletić, Grgur Matolić, Alan Ivković, Damir Hudetz, Stjepan Bulat, Frane Bukvić, Saša Janković

**Affiliations:** 1Department for Orthopaedic Surgery, University Hospital, “Sveti Duh”, Sveti Duh 64, 10000 Zagreb, Croatia; philip.vuletic@gmail.com (F.V.);; 2Faculty of Kinesiology, University of Zagreb, Horvaćanski zavoj 15, 10000 Zagreb, Croatia; 3School of Medicine, University of Zagreb, Šalata 2, 10000 Zagreb, Croatia; 4Department of Clinical Medicine, University Applied Health Sciences, Mlinarska cesta 38, 10000 Zagreb, Croatia

**Keywords:** meniscal repair, athletes, return to sport

## Abstract

Increased knowledge of the long-term destructive consequences of meniscectomy has created a shift towards operative repair of isolated meniscus lesions. However, in the literature the results of isolated meniscal repair in athletes currently remain underreported. Our objective was to investigate the clinical and functional outcomes as well as survival and return to sport in patients who underwent meniscal repair after isolated meniscal tear, with a focus on athletes (both professional and recreational) in the study population. This retrospective study included 52 athletes who underwent knee surgery for isolated meniscal tear between 2014 and 2020. Patients with concomitant ligamentous and/or chondral injury were not included in this study. The mean age of the patients was 25.5 years (ranging from 12 to 57 years). The mean follow-up period of all patients was 33.3 months (ranging 10 to 80 months). The mean purpose of the study was to report the return to sport. The International Knee Documentation Committee rating (IKDC), Lysholm score, the Knee Osteoarthritis Outcome Score (KOOS) and Tegner activity level were determined at the follow-up. Failure was defined as re-operation with meniscectomy or revision meniscal repair. In total, 44 out of 52 patients (85%) returned to their previous sports activities. At follow-up, the mean Lysholm score was 90, representing a good to excellent result. Assessment of KOOS (mean value 88.8) and IKDC (mean value 89) scores also showed good to excellent results. A mean level of Tegner scale was 6.2, indicating a relatively high level of sports participation. Failure was encountered in 8 out of 52 knees (15%). Therefore, isolated meniscal repair resulted in good to excellent knee function and most athletes can return to their previous level of sports participation.

## 1. Introduction

Both medial and lateral meniscus work together to provide kinematics required for the knee joint to function optimally. Each year over a million surgical interventions in the knee are performed due to meniscal lesions and arthroscopic meniscectomy is considered the most common surgical intervention in orthopaedic surgery [1].

As biomechanical studies have suggested, disruption of meniscal tissue affects normal joint function and alters strain on the articular surface [2]. Furthermore, the risk that the joint is prone to early degenerative changes and increased morbidity correlates even with the extent of meniscal loss after injury [3]. Thus, current scientific evidence strongly supports the fact that meniscal lesions lead to early osteoarthritic (OA) changes [4,5]. Meniscectomy is still commonly performed, however, although repair is more beneficial in terms of long-term results [6,7]. Favorable outcomes after partial meniscectomy of the knee have been demonstrated in short-term follow-up, but the long-term risk of progression of OA remains [8].

Surgical meniscus repair procedures are being increasingly performed due to accumulated knowledge of the long-term destructive consequences of meniscectomy, the benefits of meniscal preservation, and the ongoing improvements in the repair techniques and devices [9]. Secondary to the findings in the long-term follow-up, a shift in the treatment of isolated meniscus lesions has occurred, with the current gold standard being operative repair as opposed to meniscectomy [8,9]. While this is widely accepted for the general population [10,11], the treatment of such injuries in athletes remains poorly studied and controversial. Athletes with high demand on their joints and menisci consequently test the limits of the meniscal repair which potentially rises the risk of surgical failure rate. Because meniscal repair requires a longer rehabilitation period, partial meniscectomy is still routinely performed to allow a more rapid return to sports [3]. In general, good to excellent short-term results can be expected after partial meniscectomy [3]. However, not even a partial meniscectomy is without consequences for knee health in athletes. Moreover, results have shown lower incidence of return to sport after meniscectomy than reported in patients with meniscal repair due to reported persistent knee pain [3]. In contrast, a recent systematic review suggests that return to play is higher after isolated meniscus repair, with an overall rate of 83.1 percent [9].

The majority of studies that examined outcomes of meniscal repairs included meniscal repair concomitantly performed with anterior cruciate ligament (ACL) reconstruction [10]. Only some reports discussed sport-specific outcomes, such as return to sport, after isolated meniscal repair [10].

Thus, the purpose of this study was to evaluate return to sport, as well as the clinical and functional outcomes, in patients who underwent meniscal repair for isolated meniscal tear, with the focus of the study on athletes (both professional and recreational).

## 2. Methods

### 2.1. Study Design

This is a level IV case series retrospective study, which includes a case series of 52 athletes who underwent repair of isolated meniscal tear between 2014 and 2020 at the Department for Orthopaedic Surgery, University Hospital, “Sveti Duh”. Institutional review board (IRB) approval (number: 011-3342) was obtained prior to enrolment and documented informed consent was obtained from each participant. The key inclusion criteria were (1) patients with symptomatic isolated meniscal diagnosed on magnetic resonance imaging (MRI), (2) athletes who train and participate in pivoting or non-pivoting sport on daily basis either at recreational or professional level, (3) normal alignment of the tibiofemoral joint (less than 5 degrees of varus/valgus), (4) stable knee joint, (5) pre-operative MRI showing international cartilage repair society (ICRS) score less than 2 on medial or lateral compartment and confirmed at the time of surgery. Exclusion criteria were (1) athletes who do not participate in sport on daily basis, (2) patients with concomitant ligamentous and chondral injury, (5) patients with malalignment of the tibiofemoral joint, (5) local or systemic infection. Both inclusion and exclusion criteria were further confirmed intra-operatively. The outcome measure of this study was to determine the success (return to sport) and failure rate (re-tear) of arthroscopic repair for isolated meniscus tear. Furthermore, subjective patient reported outcome scores: The International Knee Documentation Committee rating (IKDC), Lysholm score, the Knee Osteoarthritis Outcome Score (KOOS) were evaluated at the last follow-up. Tegner activity level was performed in two evaluation moments: pre- and postoperative (higher score presents better outcome). Failure was defined as developing symptoms of joint pain, locking, or swelling with radiologically confirmed re-tear requiring reoperation with meniscectomy or revision meniscal repair.

### 2.2. Surgical Technique

All types of isolated meniscal tear (including radial, longitudinal (Figure 1), bucket handle and complex tears) with no other joint pathology (ligamentous or chondral lesion) were considered for repair if possible. Degenerative tears were not repaired. All repairs were performed arthroscopically with an outside-in and/or all-inside technique (Figure 2).

### 2.3. Rehabilitation

The standard post-operative regimen was conducted for 6 weeks combining protected weight-bearing with crutches and knee brace, range of motion allowing for 90° of flexion in first 6 weeks. Full weight-bearing, full range of motion and squatting followed the initial 6 week period.

### 2.4. Statistical Analysis

Statistical analysis was performed with SPSS version 24.0 for Windows (IBM SPSS Inc., New York, NY, USA). Descriptive analysis was performed for outcome variables. Basic patient characteristics, such as age, were summarized as mean, range, minimum, and maximum. Categorical variables, such as sex and type of injury, were summarized as percentages. Continuous variables were described by the mean value (± standard deviation, sd). A Wilcoxon signed rank test was used for comparison between mean scores in Tegner pre- and post-operative values. A p value of less than 0.05 with a 95% confidence interval was considered statistically significant.

## 3. Results

### 3.1. Study and Patients Characteristics

Demographic characteristics of our patient cohort are outlined in the Table 1. The mean patient age was 25.5 years (range, 12–57 years). All the patients underwent arthroscopic isolated meniscal repair. Out of 52 athletes, 38 reported non-contact type of injury. A majority of patients were men (38 out of 52) and mean BMI was 22.8. The mean follow-up period was 33.3 months (range, 10 months–80 months). No intra-operative complications and subsequent neurovascular injury or post-operative infection were reported.

### 3.2. Main Study Findings

At the last follow-up 44 patients returned to their previous sports activities at the same level as before the injury. During the follow-up period, failure (re-rupture) occurred in 8 of 52 knees (15%) with no return to the previous level of the sports activities. The mean return to sports after isolated meniscal repair was 6.5 months.

### 3.3. Subjective Outcome Scores

At the last follow-up, the mean Lysholm score was 90.3, representing a good to excellent result (Table 2). Assessment of KOOS (mean value 89) and IKDC (mean value 88.8) scores also showed good to excellent results (Table 2). The difference between pre- and post-operative mean values of Tegner score was found to be statistically significant (*p* < 0.05). A mean level of Tegner scale was 6.2 at last follow up, indicating a relatively high level of returning to sport. But there was significant decrease in Tegner score values from 7.1 pre-operative to 6.2 post-operative (Table 2).

## 4. Discussion

Increasing knowledge of the role of the menisci in loading, stability, lubrication, and proprioception of the knee joint has led surgeons to maintain and repair as much of meniscus possible [4,5]. In the literature, the functional outcomes of isolated meniscal repair in athletes currently remain underreported. Focus of this study were both high demand and recreational athletes with high demand on their joints and menisci which consequently tests the limits of the meniscal repair. Follow-up examination of the present study showed that 85% of patients returned to their previous sports activities at the same level as before the injury in an average of 6.5 months. Similarly, Logan et al. [12] reported a 5-year follow-up study of meniscal repairs in 42 elite athletes and reported that 81% of athletes returned to their preoperative athletic level after an average of 10.4 months. Those results are in accordance with the data from literature showing high return rate to sport of 86% in athletes and 90% in the general population [8]. Some surgeons prefer partial meniscectomy, which is particularly advantageous in the medial compartment because it allows for a shorter rehabilitation protocol, earlier return to sports, and good to excellent short-term results after partial meniscectomy [3,13]. Because of the established biomechanical role of the menisci in the knee joint, meniscectomy in patients with high functional demands may accelerate early degenerative changes, leading to decreased performance and possibly a shorter career [13,14].

In a study of a case series of 46 patients who underwent repair of isolated meniscal tear, a failure rate of 8.7% was noted [5]. However, these results could be influenced and underestimated due to the short follow-up period (average 19.8 months) [5]. In studies with longer follow-up (more than 20 months) after meniscal repairs for isolated meniscal tear, higher rates of re-tear of 23.7% [15] and 26.9% [3] were found.

In the literature, the effects of concomitant ACL reconstruction on meniscal repair are inconsistent. Higher meniscal healing rates and better outcomes in knees with ACL reconstruction have been associated with increased joint blood flow after surgery and the more peripheral and vertical orientation of meniscal tears in ACL injuries [16]. In contrast to this assumption, recent studies have shown similar success rates for isolated meniscal repairs compared with those with concurrent ACL reconstruction [17,18].

According to Blanchard et al. [9], the two subtypes of arthroscopic techniques had very similar return to play rates of 95.5% for the all-inside technique and 91.4% for the inside-out approach. With the advent of reliable all-inside techniques for the most central parts of the posterior horn tears, combined type of arthroscopic suture techniques for complex meniscal tears is becoming feasible.

Meniscal repair for isolated meniscal tear has been shown to result in better patient-specific outcomes and less cartilage degeneration than partial meniscectomy [10]. Our results in terms of functional scores (KOOS, IKDC, Lysholm) were at levels considered good to excellent. Comparable data were noted between pre-injury and reported post-operative Tegner scores reflected by the high return to sport rate of 85%.

However, we must note that limitation of this study was that there was no control group, and that the success of the procedure was based on the indirect evidence of meniscal healing as patient being asymptomatic. Our study did not include a detailed description of athletic activity, so the results could not be analyzed in relation to the different disciplines and levels of athletic activity. Future studies need to include a larger number of cases to clarify the above points. In addition, a longer follow-up period with clinical and radiological examinations is needed to describe the beneficial effect of meniscal repair on delaying the progression of OA in high-level athletes.

This study presents the fourth largest study evaluating clinical outcomes of arthroscopic repair for isolated meniscus tear in athletes [9]. Eggli et al. [3] and Eberbach et al. [12] have proven that isolated meniscal lesion repair provides good to excellent results in patient reported outcome score with a high level of returning to previous sports activities. Good clinical results and reported safety of the procedure demonstrates the success of the isolated repair of meniscal tear as a surgical option, especially in young and active patients.

## 5. Conclusions

In athletes with isolated meniscus tear, isolated meniscal repair resulted in good to excellent knee function and high percentage of returning to sport at the same level as before injury. During the follow-up period ranging from 10 to 80 months (mean, 33.3 months), failure of the repaired site was encountered in 8 of the 52 knees (15%).

## Figures and Tables

**Figure 1 ijerph-20-05088-f001:**
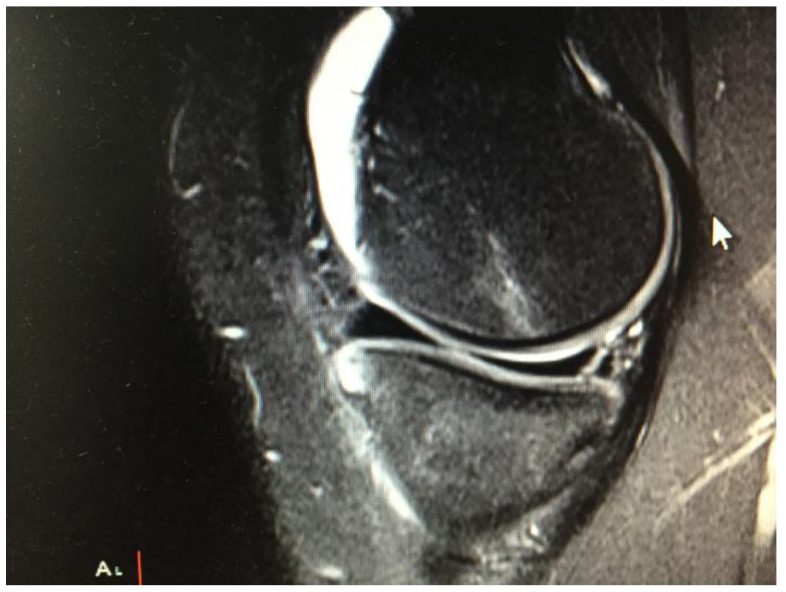
Magnetic resonance (MR) sagittal view of a right knee in 20-year-old male patient with a longitudinal meniscal tear in posterior part of medial meniscus.

**Figure 2 ijerph-20-05088-f002:**
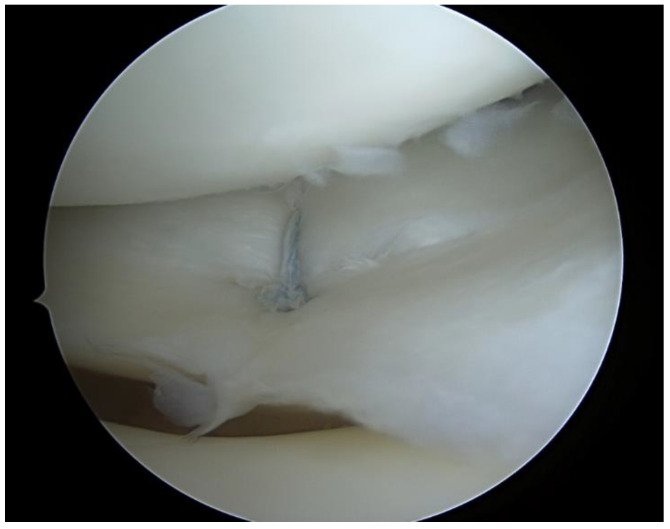
Arthroscopic image demonstrating an all-inside meniscal repair of medial meniscus.

**Table 1 ijerph-20-05088-t001:** Demographics of patients.

Age, (years)	25.5 (range, 12–57)
Male:female, n	38:14
Type of injury, n (contact:non-contact)	14:38
Follow-up period (months)	33.3 (range, 10–80)

**Table 2 ijerph-20-05088-t002:** Summary of patient reported outcome scores.

	Pre-Operative Tegner	Post-Operative Tegner	Post-Operative Lysholm	Post-Operative ^†^ KOOS	Post-Operative ^‡^ IKDC
Mean (±sd)	7.1 (±2.1)	6.2 (±2.2)	90.3 (±10.7)	89 (±11.1)	88.8 (±17.9)

^†^ KOOS—the Knee Osteoarthritis Outcome Score; ^‡^ IKDC—the International Knee Documentation Committee rating.

## Data Availability

Not applicable.

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
