# Peer review of "Clinical Outcome of Arthroscopic Repair for Isolated Meniscus Tear in Athletes"

_ijerph, 2023, doi:10.3390/ijerph20065088_

Round 1

Reviewer 1 Report

The manuscript attempts to present the clinical and functional outcomes as well as survival and return to sport in patients who underwent meniscal repair after isolated meniscal tear, with a focus on 52 athletes (both professional and recreational) in the study population (2014 and 2020). The paper is very interesting with explanations of the importance and details, and the results of isolated meniscal repair in athletes currently remain underreported in the literature. The repairs were performed arthroscopically with an outside-in and/or all-inside technique. Besides this, the authors showed that 85% of patients returned to their previous sports activities at the same level as before the injury in quite a short period. The failure of the repaired site was encountered only in 8 of the 52 knees. The experience of the authors with this procedure can offer new studies for other scientists in the future.

I would like to suggest some minor corrections for a better quality of this manuscript from my side and the most important details I have underlined in the comments. 

General comments

Point 1: What are the future directions of this study? This point of view is very important and must be explained in the general text.

Point 2: I suggest to the authors add more in the text and in the references list publications and especially from the last 2 years too because there were not any references from these periods. Other references comprise information from quite old periods.

Point 3: Please, improve the list of references, according to the guidelines of them for the authors.

Specific comments

Point 1: Please, look at the text in Line 22: “The International Knee Documentation Committee” … and there add the abbreviation (IKDC) like it is the general text of the article in Line 82.

Point 2: Please, in Line 63 add a complete explanation of the abbreviation ACL because some readers could not understand the meaning of it too.

Point 3: Please, try to use one size of the Figures, and the quality of Figure 1. could be better in a lesser size of it.

Point 4: Please, try to use one style (comma or semicolon) in the text and compare text (range, 12-57 years) in Line 115 and (range; 10 months-80 months) in Line 118. Maybe there can be better used the period in years instead of months?

Point 5: Please, compare the data of the text in Line 132 and Table 2. The mean values of KOOS and IKDC in the text are different (opposite) from these mean values in Table 2.

Point 6: Please, add punctuation after the author in the text “… al” in Line 163.

Author Response

Dear Reviewer, we kindly provide You with point-by-point response.

Point 1:  What are the future directions of this study? This point of view is very important and must be explained in the general text.

Response to point 1: Corrected in manusript.

„Our study did not include a detailed description of athletic activity, so the results could not be analyzed in relation to the different disciplines and levels of athletic activity. Future studies need to include a larger number of cases to clarify the above points. In addition, a longer follow-up period with clinical and radiological examinations is needed to describe the beneficial effect of meniscal repair on delaying the progression of OA in high-level athletes.“

Point 2: I suggest to the authors add more in the text and in the references list publications and especially from the last 2 years too because there were not any references from these periods. Other references comprise information from quite old periods.

Respones to Point 2: Corrected in manuscript.

Point 3: Please, improve the list of references, according to the guidelines of them for the authors.

Response to Point 3: Corrected in manuscript.

Specific comments

Point 1: Please, look at the text in Line 22: “The International Knee Documentation Committee” … and there add the abbreviation (IKDC) like it is the general text of the article in Line 82.

Response: Corrected in manuscript.

Point 2: Please, in Line 63 add a complete explanation of the abbreviation ACL because some readers could not understand the meaning of it too.

Response: Corrected in manuscript.

Point 3: Please, try to use one size of the Figures, and the quality of Figure 1. could be better in a lesser size of it.

Response: Corrected in manuscript.

Point 4: Please, try to use one style (comma or semicolon) in the text and compare text (range, 12-57 years) in Line 115 and (range; 10 months-80 months) in Line 118. Maybe there can be better used the period in years instead of months?

Response: Corrected in manuscript.

Point 5: Please, compare the data of the text in Line 132 and Table 2. The mean values of KOOS and IKDC in the text are different (opposite) from these mean values in Table 2.

Response: Corrected in manuscript.

Point 6: Please, add punctuation after the author in the text “… al” in Line 163.

Response: Corrected in manuscript.

Reviewer 2 Report

It was a pleasure to read the article and make my comments. Knee injuries are worrisome.

Abstract

Presents structured summary

Introduction

The researcher contextualizes and presents the existence of studies on the subject. It also presents the gap in the literature. Introduce the objective.

Methodology

Present the study design

Please provide Research Ethics Board approval number

Inclusion and exclusion criteria

Assessment instruments

-       The International Knee Documentation Committee rating (IKDC)

-       Lysholm score,

-       Knee Osteoarthritis Outcome Score (KOOS)

-       Tegner activity level were evaluated at last follow-up

-       Failure was defined as developing symptoms of joint pain, locking, or swelling with radiologically confirmed re-tear requiring reoperation with meniscectomy or revision meniscal repair.

The researcher must describe that the International Knee Documentation Committee rating (IKDC), Lysholm score and Knee Osteoarthritis Outcome Score (KOOS) was performed at a time of assessment (follow-up). The researcher should describe in more detail about each of these assessment instruments.

The Tegner activity level was performed in two evaluation moments (pre and postoperative), therefore, it must be described in the methodology. The score of this test should be described (eg, higher is worse?)

Was a normality test performed for the Tegner activity level?

Discussion

The authors discuss their results with the literature

Presents the limitations of the study

The researchers could present what the study brings to the scientific community and what the future perspectives are

Author Response

Dear Reviewer,

we thank You for taking Your time and we kindly appreciate Your comments.

We have presented our methods more adequately now with all the inclusion and exclusion criteria.

The Lysholm Knee Score is a disease-specific outcome measure that includes eight domains: Limping, Locking, Pain, Stair Climbing, Use of Supports, Instability, Swelling and Squatting. A total score of 0 to 100 points is calculated, with 95 to 100 points representing an excellent outcome, 84 to 94 points representing a good outcome, 65 to 83 points representing an adequate outcome, and < 65 representing a poor outcome. The Tegner Activity Scale is a numerical scale that ranges from 0 to 10. Each score indicates the ability to perform certain activities. An activity level of 10 corresponds to participation in competitive sports, including soccer, football, and rugby at the elite level; an activity level of 6 points corresponds to participation in recreational sports; and an activity level of 0 is assigned if a person is on sick leave or receiving a disability pension because of knee problems.

Use of the Lysholm score, the Tegner activity scale, KOOS, IKDC provide a comprehensive and (literature) validated outcome assessment for patients with meniscal injuries and meniscal procedures (both meniscectomy and repair).

Sincerely Yours